# Effects of Static Icing on Flashover Characteristics of High-Speed Train Roof Insulators

**Zhibin Qi** [1]**, Ruiping Zhang** [1]**, Jianqiao Ma** [1,]*****, Xiangfei Wang** [1]**, Lei Ma** [1]**, Zongyu Zhao** [2] **and Youqiang Wu** [3]

[1] Department of Power Engineering, School of Automation and Electrical Engineering, Lanzhou Jiaotong University, Lanzhou 730070, China; lzjtqzb@126.com (Z.Q.); zhrp@mail.lzjtu.cn (R.Z.); wxfdbljx@126.com (X.W.); lzjtml@163.com (L.M.)
[2] China Railway Lanzhou Group Co., Ltd., Lanzhou 730070, China; helen_lixiaojia@163.com
[3] Wenzhou Yikun Electric Co., Ltd., Wenzhou 325401, China; yikun1693@126.com
***** Correspondence: lzjdmjq@163.com; Tel.: +86-189-0931-0132

**Abstract:** To study the icing flashover characteristics displayed by insulators on the roof of a high-speed train at a low-speed or static state, an environment with a low temperature and high humidity is established in the artificial climate chamber to carry out the icing flashover test of composite insulator under a static state. The results of the study indicate the presence of ice ridges bridging the sheath at the edge of the sheath when the ambient wind speed is less than 2m/s. The arc appears firstly at the end of the ice ridge, and it is likely that the arc at the adjacent tip converges in the air and then further extends to the edge of the next sheath. The arc melts the ice ridges at the end and enables it to stick to the tip of the ice ridge in the form of water droplets, and therefore the existence of droplets results in the distortion of the electric field near them. Moreover, the elongation of droplets and the bridging of ice edges accelerate the process of flashover. The research results can offer a reference to the structural optimization design of anti-icing insulators for high-speed trains in the future.

**Keywords:** high-speed train; insulator on the roof; static icing; flashover characteristics; distortion of electric field



## 1. Introduction

As of December 2020, the operating mileage of high-speed rail in China has reached 38,000 km. The meteorological environment along the high-speed railway is complex and changeable, and the meteorological conditions such as rain, frost, and fog, may cause abnormal flashover to the insulators on the train roof [1–4].

In cold weather, fog in the air may form condensation, frost, ice layer or ice ridge on the surface of roof insulator, which will cause the decrease of surface insulation resistance of roof insulator and the increase of surface leakage current [5–7]. When the phenomenon of severe icing existing on the edge of the sheath occurs, it is likely that the sheaths may be bridged by the ice ridges, and the creepage distance of the insulator after the bridging will be shortened, which will result in the fact that the local air gap will withstand a higher voltage. When the voltage at both ends of the air gap goes beyond its bearing capacity, the air gap will be broken down, and in a serious situation, the accident of insulator ice flashover will be directly triggered [8,9]. The degree to which the extent of the surface of the sheath is iced directly affects the insulation characteristics shown by the insulator, and the degree of icing is usually represented by parameters including the thickness of the icing, the diameter of the ice ridges, and the length of the ice ridges, etc. The changes existing in the three parameters have a close relationship with the factors including the material of the insulator sheath, the sheath protrusion, the diameter of the sheath, the number of water droplets, the diameter of the water droplets, and the wind speed [10–13], etc. The analysis on the movement trajectory of water droplets circling around the insulator can

be carried out by adopting the method of flowing around the cylinder. With regard to the amount of ice coating on the surface of the sheath for the roof insulator, it is determined by the parameters such as collision rate of water droplet, capture rate of water droplet, and freezing rate of water droplet, which have close relationship with airflow velocity [14,15]. Under such circumstances, the roof insulator is in a static or low-speed state, and it is conducive to the formation of ice. The ice formation, ice melting and flashover process under the circumstance of static or low-speed airflow function as the basis for the analysis on the ice-covered flashover characteristics of the roof insulator in the environment where the action of high-speed airflow is available. During the period when the train stops at the station or intersection, waiting for the dispatch instructions, the icing environment of the roof insulators are mainly low temperatures, high humidity, and low-speed airflow. It is acknowledged that the formation of ice ridges is easier in the static or low-speed airflow environment. After the bridging of ice ridges, the creepage distance of the insulators decreases, and it serves as a greater threat to the reliable operation of the roof insulators. A strong field area is available at the end of the ice ridges, and it is in the area that the phenomenon of discharge will occur. When there are water droplets hanging at the end of the ice ridge, the possibility of discharge at the end of the ice ridge will witness a sharp increase due to the water droplets with high conductivity, and the probability of flashover will see an increase as well [16–18].

In comparison with the insulators in the power system, the roof insulators of electric locomotives are small in volume, and the number of units conducting studies on the characteristic of insulation shown by the roof insulators is relatively small. Existing literature related to the characteristics shown by the ice flashover of roof insulators mainly attaches importance to the description over the phenomenon of the ice flashover itself and the qualitative analysis on the causes, instead of offering a systematic analysis and implementation plan of practical engineering. Therefore, it is necessary to carry out a systematic analysis on the characteristics shown by the ice flashover of roof insulators. The ice cover test in this paper is carried out in a static or low velocity airflow, low temperature and high humidity environment. Regarding the formation of roof insulators and the characteristics shown by the flashing of melting ice with the action of high-speed airflow, follow-up studies will be carried out in a direct-blown wind tunnel where there will be at a maximum airflow speed of 80 m/s.

This paper is intended to analyze the influence of static icing on the flashover characteristics shown by the insulators of a high-speed train roof through the combination of experiment and simulation. First of all, in the artificial climate chamber, the test of ice-covered silicone rubber composite insulator on a high-voltage isolation switch used in high-speed trains is carried out under the working conditions, where the wind speed is less than 2 m/s, and then the development process of the flashover arc under severe icing conditions such as ice ridge bridging is obtained using the method of continuous boost. Based on the parameters such as the appearance of icing in an icing test and the test of icing flashover, a three-dimensional model is established to simulate and analyze the distortion characteristics of local electric field strength caused by the parameters, such as the length of the ice ridge, the position of ice ridge, and hanging water droplets. The results obtained from the simulation can offer a reference to the selection of the sheath spacing and the value of sheath extension in the insulator structure design of the high-voltage isolation switch on the roof of the high-speed train.

## 2. Icing Test

### 2.1. Specimens

As shown in Figure 1, there is the appearance of the hard silicone rubber composite insulator of the high-voltage isolation switch on the roof of a high-speed train. The sheath arrangement is alternating between large and small sheathes, with a total of nine pieces numbered 1#~9#, and the parameters of the insulator structure are shown in Table 1.

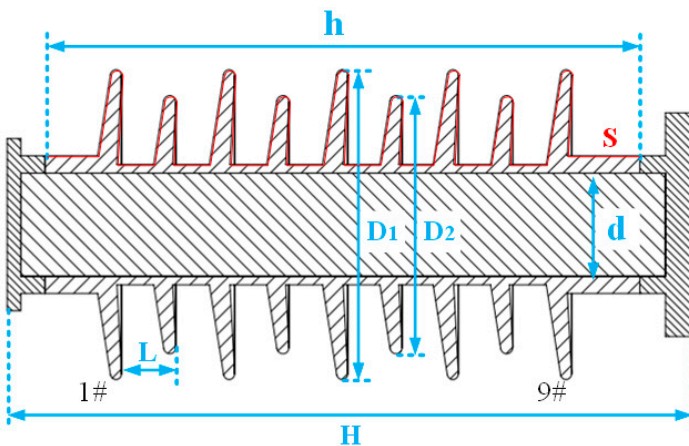

**Figure 1.** Profiles of the composite insulator used.

**Table 1.** Structural parameters of silicone rubber insulator on the roof of a high-speed train.

| Type | Parameter (mm) |
|---|---|
| Structure height (H) | 400 |
| Insulation height (h) | 350 |
| Sheath diameter (D1/D2) | 180/150 |
| Mandrel diameter (d) | 60 |
| Creepage distance (s) | 1140 |
| Sheath spacing (L) | 32 |

*2.2. Test Facilities*

The tests of insulator icing and flashover tests are carried out in an artificial climate chamber, with the dimension of 4.5 m × 3.7 m × 2.9 m. With two sets of refrigerating units installed in the climate chamber, the indoor temperature can be adjusted to as low as −20 °C. In the climate chamber, the walls and top are filled with polyurethane sheets with a thickness of 150 mm, and the stainless steel, with a thickness of 0.8 mm, is equipped on the wall surface.

With adjustable pressure of the water pump, the sprinkler system is composed of four parts—water pump, pipeline, nozzle, and bracket. The nozzles employed in the test of the insulator icing are standard IEC nozzles. By changing the water pump pressure and the pressure of the air compressor, the droplet size and mass flow rate can be changed. The test device used in the icing test satisfies the requirements of relevant specifications [19].

With the rated output voltage of the transformer used in the power frequency test as 150 kV, the test power supply is 50 Hz AC power supply, the rated output current as 1.5 A, and the voltage divider ratio as 1000:1, the parameters employed in the power supply of the test meet the requirements for the ice-covered insulator of the flashover test [19,20]. The appearance of the artificial climate chamber and the transformer used in the power frequency test is shown in Figure 2, and the wiring principle of the test is shown in Figure 3.

*2.3. Test Procedures*

The test of the uncharged insulator and flashover test are carried out in the artificial climate chamber. With the pre-cooling water sent to the nozzle through the pump and pipe, a bracket is placed symmetrically on the left and right sides of the insulator. Moreover, two nozzles are arranged on the higher and lower position in a fixed pattern, respectively. The vertical spacing of the nozzle is 600 mm, and the distance between the nozzle and the insulator is 1000 m. After several tests on the nozzle angle, the spray device can freeze the three insulators at the same time.

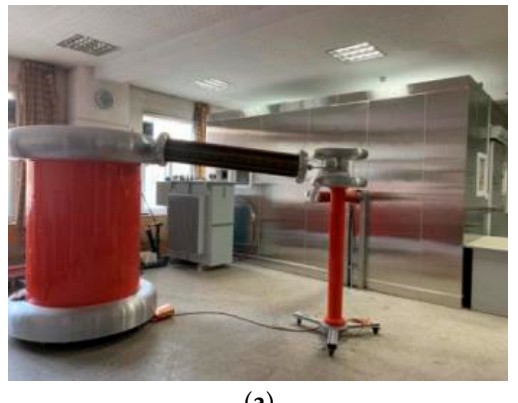 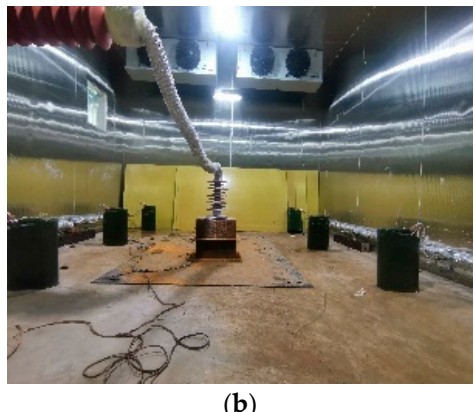

|(**a**)|(**b**)|

**Figure 2.** (**a**) Test transformer; (**b**) artificial climate chamber.

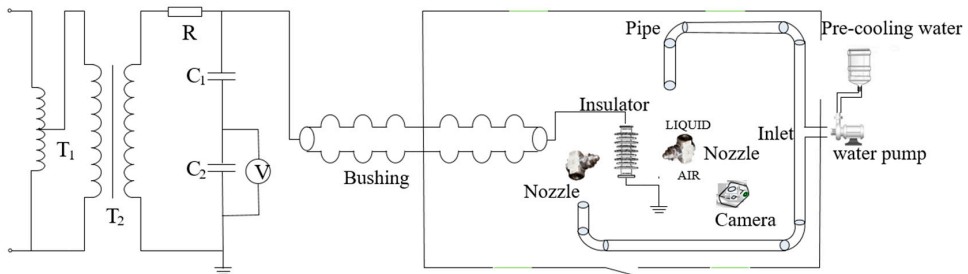

T1-Voltage regulator; T2-Transformer; R-Protective resistor;

$C_1$ and $C_2$-High voltage arm capacitance; V-Voltmeter

**Figure 3.** Schematic diagram of test wiring.

Before the test, clean the insulator of the test sample, and dry it in the shade, before pre-cooling the ice-coated water to the temperature of 3~4 °C. After decreasing the temperature of the climate chamber to −10 °C, put two clean insulators in it and connect the test line, followed by spraying the pre-cooling water through the spray device onto the surface of the insulator, and begin covering the ice. During the entire process of icing, maintain the indoor temperature of the climate at −7~−8 °C, and the relative humidity at 95%~100%. Then, record the morphology of the insulator icing every 10 min, including the location where the ice ridge is distributed, the length of the ice ridge, and the diameter of the ice ridge, etc., for the convenience of subsequent modeling. When some ice ridges bridge the sheaths, stop spraying and continue freezing for 15 min, enabling the ice surface to freeze completely.

Carry out the flashover test of the ice-covered insulator and photograph the development process of the flashover arc using the method of continuous boost. During the process of flashover, the hanging water droplets at the end of the ice ridge serve as one of the basis for the subsequent establishment of the three-dimensional model.

## 3. Experimental Results

### 3.1. Icing Morphology

Carry out the icing test in accordance with the test method mentioned above in Section 2.3. During the test, the test water was tap water, and its conductivity was measured to be 350–420 μs/cm. Ice layer and ice ridge on the surface of sheath as shown in Figure 4.

It can be obtained from observation that the upper surface of the insulator sheath is wrapped in a thin layer of ice, in terms of the ice-clad morphology of the insulator, and the thickness of the ice layer gradually increases from the root of the sheath to the edge, with the average thickness of the ice layer about two millimeters. Moreover, the whole surface of the ice is relatively smooth, with local bumps, which are mostly distributed on the ice

surface directly below the ice ridge on the edge of the sheath. Basically with no ice covered on the lower surface of the sheath, there are scattered ice droplets attached.

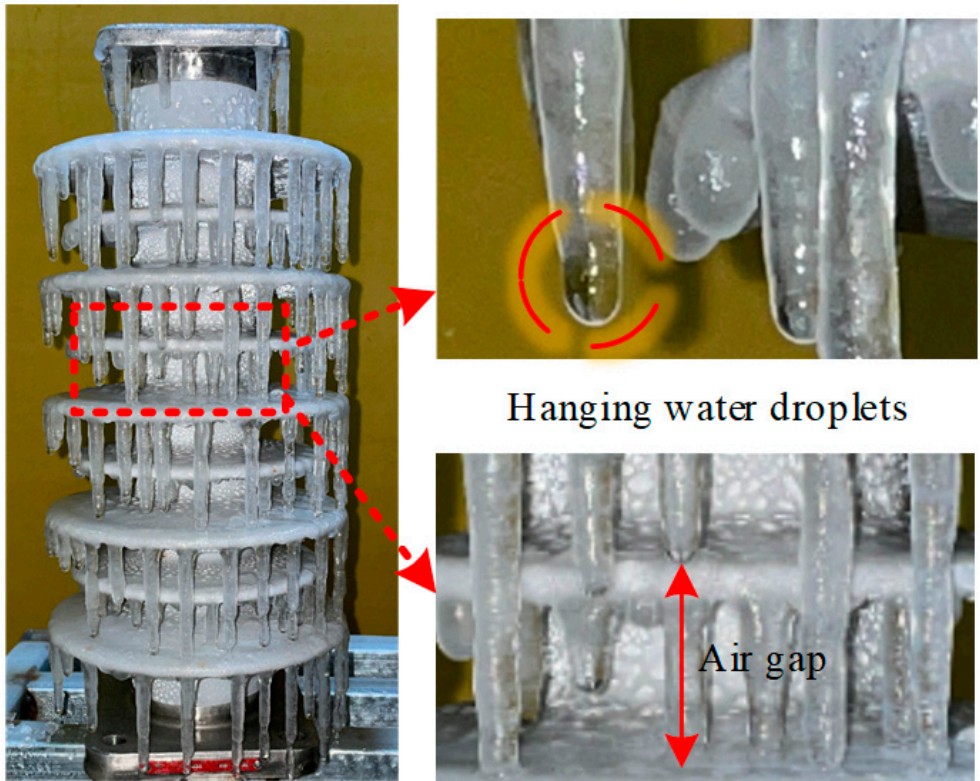

**Figure 4.** Icing topography.

Hanging from the edge of the large sheath are a large amount of ice ridges, which are evenly distributed and located slightly closer to the inner side of the sheath edge. In addition to some ice ridges, which bridge the sheaths, there are ice ridges with lengths mostly between 30 and 50 mm. In the follow-up simulation modeling, the parameters of the lengths for ice were selected as 30, 40, and 50 mm. Vertical and smooth, the ice ridges were free of bubbles and no obvious change was available in the diameters at the root of the ice ridges. With a length of about seven millimeters, the diameters at the tips are within the range of two to four millimeters. In comparison with the large sheath, the degree of icing on the small sheaths is significantly lower, and the ice ridges bridge sheaths partially. With regard to the number of bridges and the length of the ice ridges, they witness a significant reduction.

*3.2. Ice Flashover Test*

The flashover test of the insulator was carried out using the method of continuous boost. The continuous voltage boost method is to wire the ice-covered insulator in accordance with the flashover test wiring schematic, check the wiring is correct, and then increase the voltage at a certain rate until the test product flashover.

The process of arc development during the process of discharge is shown in Figure 5, Figure 6 shows the local arc, and the morphology of the ice layer on the surface of insulator after flashover is shown in Figure 7.

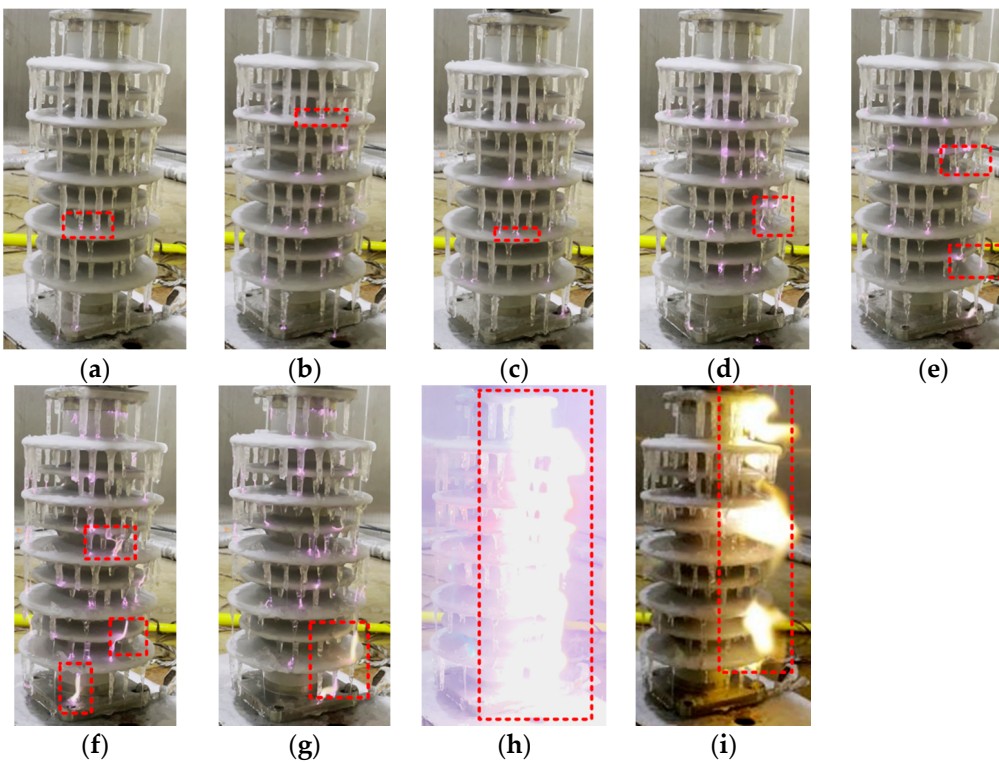

**Figure 5.** Icing flashover process. (**a–i**): please add explanation here. (**a,b**) Initial stage; (**c,d**) The stage of corona discharge; (**e,f**) The stage of local arc development; (**g,h,i**) Continuous development of the arc until the flashover stage.

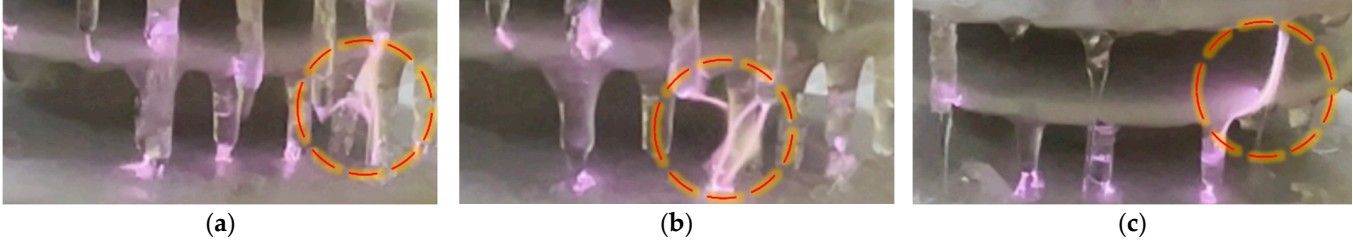

**Figure 6.** Local arc: (**a**) Ice tip arc; (**b**) Splitting arc; (**c**) Ice melting water arc.

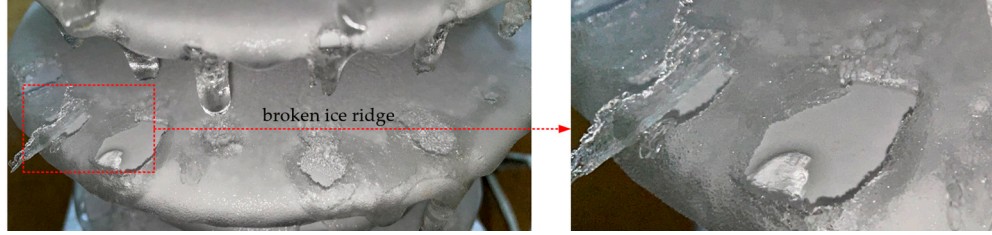

**Figure 7.** Ice morphology on insulator surface after ice melting and flashover.

By observing the process of discharge in Figure 5, the ice-covered flashover of the insulator can be divided into the following four stages:

1. Initial stage (Figure 5a,b)

Having been covered with ice, the insulator has its flashover channel as the combination between "ice ridge–sheath" and air gap. At this stage the voltage is low, the surface of the ice-covered insulator is drier and the resistance of the ice is higher. Sporadic purple arcs, which are short and small, are only available at the high-voltage end and the low-voltage

end. Due to the problem about the zero-crossing phenomenon of the AC voltage, a process of extinguishing and re-ignition exists in the arc.

2.    The stage of corona discharge (Figure 5c,d)

Along with the increase in voltage, the ice layer gradually melts, and the resistance of the ice layer decreases. Moreover, the purple arc at the tip of the ice edge undergoes rapid development in a split-like manner (as shown in Figure 6). With relatively random arc development, and the ice surface under the ice ridge directly melts rapidly, and the corona noise of sizzle can be heard.

3.    The stage of local arc development (Figure 5e,f)

When the melting of the ice layer on the surface of the insulator quickens its speed, the water of melted ice continues to drip along the ice edge, and then the path of local arc develops from the tip of ice to the surface of sheath along with the running of water droplets, when the process of arc re-ignition slows down. This is due to the continuous drop of melting ice water reduces the energy of the arc, resulting in partial arc extinction.

4.    Continuous development of the arc until the flashover stage (Figure 5g–i)

The arc undergoes violent development, when the arc foot drifts to a great extent. Due to the discharge in the middle of the ice ridge at the edge of the small sheath, the ice ridge is broken, and then the water of melted ice flows down along the ice ridge (Figure 6c). For the reason that the conductivity of the water from melting ice is much greater than that of the ice ridges, the continuous flow of ice-melting water provides the development of the arc with an ideal path, and the development of arc continues along with the water flow. The process "establishment, extinction, and re-ignition" of the arc repeats itself until it penetrates into both the high and low-voltage electrodes of the insulator.

It can be observed from Figure 7 that after the completion of the flashover, the ice layer on the surface of the insulator under the ice edge melts completely, but there is a remaining ice layer in a small amount in other parts, and broken ice ridges are available on the edge of the sheath.

## 4. Analysis of Influence of Ice Ridge on Electric Field Distribution

The end of the ice ridge leads to the uneven distribution of electric field in the air domain nearby, and different distribution of field strength will influence the path and speed of the arc development. The high temperature in the process of arc development will melt part of the ice layer into water droplets, which will hang on the tip of the ice ridge, and then result in the serious distortion of the electric field near the tip of the ice ridge. Simulated analysis is conducted to figure out the influence of the changes in parameter, such as the length of the ice ridge, position of the ice ridge, and water droplets hanging at the end of the ice ridge, on the distribution of electric field intensity in the air domain nearby.

### 4.1. Simulation Model

The finite element simulation software COMSOL is used to simulate the insulator electric field. The ice-covered insulator electric field simulation model is shown in Figure 8, where Figure 8a is the air domain with its surface set to zero potential; Figure 8b is a 1:1 insulator three-dimensional ice-covered model built based on the insulator structural parameters and ice-covered morphology in Section 2.1. It is specified that the marked side at the nameplate of the insulator is the front reference plane, and the surface of the sheaths 1#, 3# and, 5# and the position on the right side of the ice ridge are selected as the research objects, as shown in Figure 8c. The simulation parameters are displayed in Table 2.

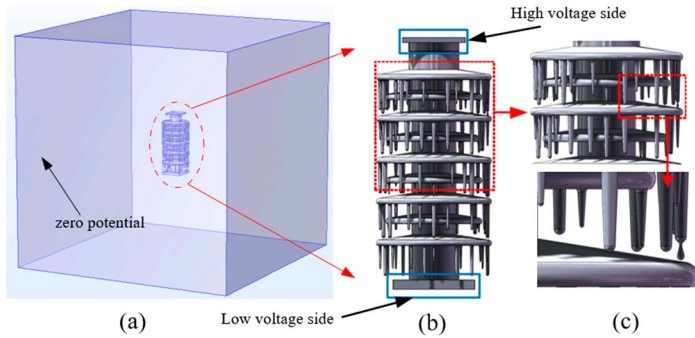

**Figure 8.** Electric field simulation model of the ice-covered insulator (**a**–**c**).

**Table 2.** Simulation parameters setting.

| Type | Relative Permittivity | Conductivity (μs/cm) |
| --- | --- | --- |
| Ice | 75 | $10^{-6}$ |
| Air | 1.01 | $10^{-12}$ |
| Silicone rubber | 4.2 | $10^{-12}$ |
| Water film | 81 | 300 |
| Mandrel | 8 | $10^{-10}$ |

Due to the complex shape of the insulator itself, as well as the need for better mesh quality of ice edges and water droplets when establishing the icing model, tetrahedral elements are selected to divide the mesh. A tetrahedral unit can better deal with the surface bending and tiny slit of the insulator, and densify at the ice edge and water drop. Select four grid numbers as shown in Table 3 below:

**Table 3.** Grid quality information under different grid numbers.

| Type | Grid 1 | Grid 2 | Grid 3 | Grid 4 |
| --- | --- | --- | --- | --- |
| Number of meshes (million) | 3.5 | 13.14 | 39.2 | 51.29 |
| Minimum mesh size (mm) | 2.25 | 1 | 0.3 | 0.2 |
| Average unit quality | 0.6575 | 0.6617 | 0.6548 | 0.6519 |
| Minimum unit quality | 0.01 | 0.01 | 0.0047 | 0.0026 |

As can be seen from Table 3, the increase in the number of grids has little effect on the cell quality, and the grid number of 39.2 million with a minimum grid size of 0.3 mm is selected for the subsequent simulation calculation by integrating the computer resources and the calculation accuracy. The ice layers on the sheaths 1#, 3#, and 5# and the mesh on ice ridge on the right side are finely divided, as shown in Figure 9. The high voltage side of the insulator is set to 38,890 V, while the low voltage side and the outer surface of the air bag are set to the potential of zero.

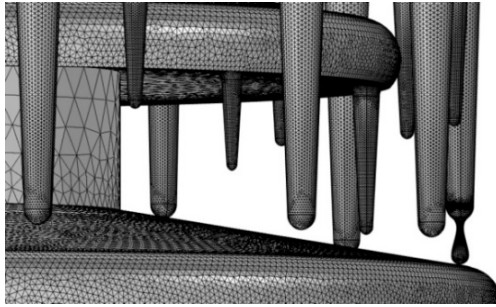

**Figure 9.** Mesh generation.

### 4.2. Influence of Ice Ridge Length on Electric Field Distribution

With the distribution of the electric field on the surface of the clean (ice-free) insulator as a benchmark, analysis on the influence of ice ridges with different lengths on the electric field distribution in the space nearby in sheath 1# is conducted using the electric field nephogram and the three-dimensional cross-section method. As shown in Figure 10, in the schematic diagram of the three-dimensional cutting line, the line segments AB in the figure refer to the lengths of the ice ridges, and the values are chosen as zero millimeters (clean insulator), 30, 40, and 50 mm, respectively. When the length of the ice ridge is 50 mm, the point one millimeter from the end of the ice ridge below the end of the ice ridge is defined as point C, that is, the length of AC is 51 mm. The longer the ice ridge, the closer the distance from point B to point C. In the follow-up simulation, the data related to the space electric field of cutting line segment CD is extracted to analyze the effects of length change in the segment CD on the distortion of the electric field. Figure 10 shows the cloud diagram of the electric field distribution under the ice ridges of different lengths, and Figure 11 shows the effect imposed by the changes of parameters, the ice length AB, on the electric field distortion in the segment CD.

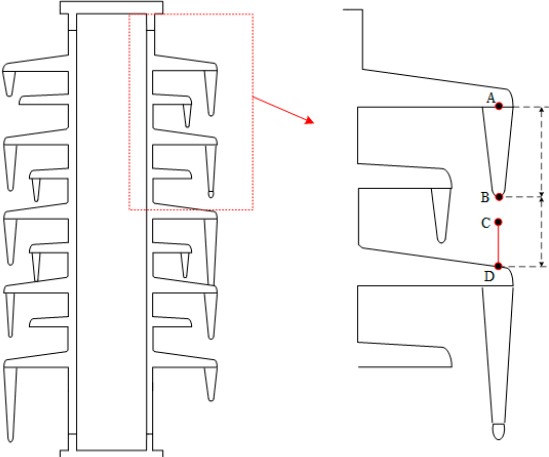

**Figure 10.** Schematic diagram of ice ridge and air gap.

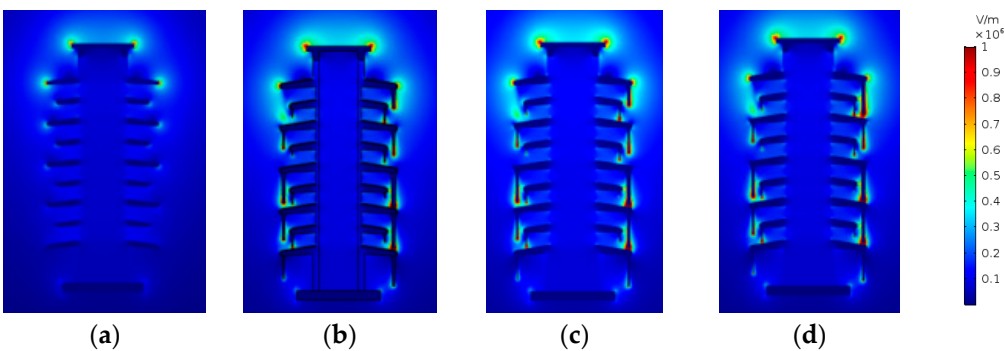

**Figure 11.** Cloud chart of electric field distribution of insulators with different ice ridges. (**a**) Ice ridge length 0mm; (**b**) Ice ridge length 30mm; (**c**) Ice ridge length 40mm; (**d**) Ice ridge length 50mm.

It can be seen from Figure 11 that obvious distortion is available in the electric field near the clean insulator and at the edge of sheath 1#, and the degree of distortion in the electric field is alleviated successively. There is the distortion in the electric field near the ice ridge, and the distortion in the electric field at the tip of the ice ridge reaches the maximum. It can be seen from Figure 12 that along with the increased length of the ice ridge, the distortion of the electric field in the fixed air domain under the ice ridge tends to

be more and more obvious—the longer the length of the ice ridge, the higher the degree of distortion, which is consistent with the actual situation.

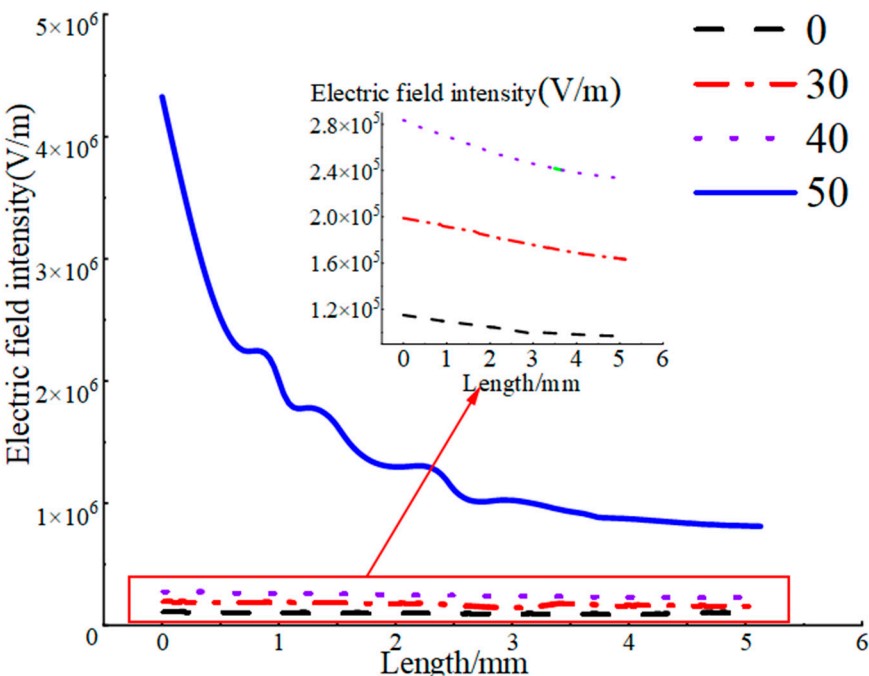

**Figure 12.** The influence of the length of the ice ridge on the electric field intensity.

Taking the average electric field strength and the distortion rate of electric field strength as the evaluation criteria, the potential difference of BD section is extracted when the ice ridge length AB is 0 mm, 30 mm, 40 mm and 50 mm, then the average electric field strength of BD is calculated, as shown in Table 4. The electric field strength distortion rate is calculated according to Equation (1) [21].

$$\eta = \frac{E_{av} - E_{jj}}{E_{jj}} \tag{1}$$

where $\eta$ refers to the distortion rate of field strength; $E_{av}$ denotes the average field strength of the ice-covered insulator corresponding to the gap BD; $E_{jj}$ is the average field strength of the clean insulator in the circumstance of corresponding air gap. $E_{jj}$ is obtained through calculation after extracting the potential difference of the line segment BD in the simulation model of the clean insulator. As point A is fixed, and the coordinates of point B change along with the changes in the length of AB, the values of $E_{jj}$ are subjected to changes along with the changes in the lengths of AB.

**Table 4.** Air gap field strength of Iced Insulator.

| Type | Condition 1 | Condition 2 | Condition 3 | Condition 4 |
|------|-------------|-------------|-------------|-------------|
| AB | 0 | 30 | 40 | 50 |
| BD (mm) | 57 | 27 | 17 | 7 |
| $E_{jj}$ (kV/mm) | 0.146 | 0.061 | 0.033 | 0.02 |
| $E_{av}$ (kV/mm) | 0.146 | 0.2987 | 0.472 | 1.139 |
| $\eta$ (%) | 0 | 389 | 1330 | 5595 |

It can be seen from Table 4 that, the length of the air gap decreases and the average field intensity increases along with the changes in the lengths of the ice ridge. In comparison with the clean insulator, the distortion rate about the intensity of average field reaches as high as 5595% when the length of the ice ridge is 50 mm.

### 4.3. Influence of Ice Ridge Bridging on Electric Field Distribution

To study the influence imposed by ice ridge-bridged sheaths on the distribution of electric field near iced insulator, a three-dimensional section is drawn, as shown in Figure 13. With the total length of the sectional line as 152 mm, the section AB refers to the ice ridge, with a length of 57 mm, which bridges the sheaths. The influence imposed by the changes in the length of ice ridges on the electric field of the three-dimensional section line is shown in Figure 14.

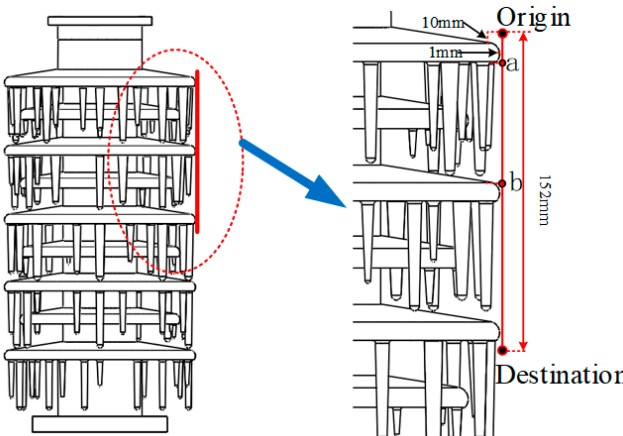

**Figure 13.** Sketch map of 3D section.

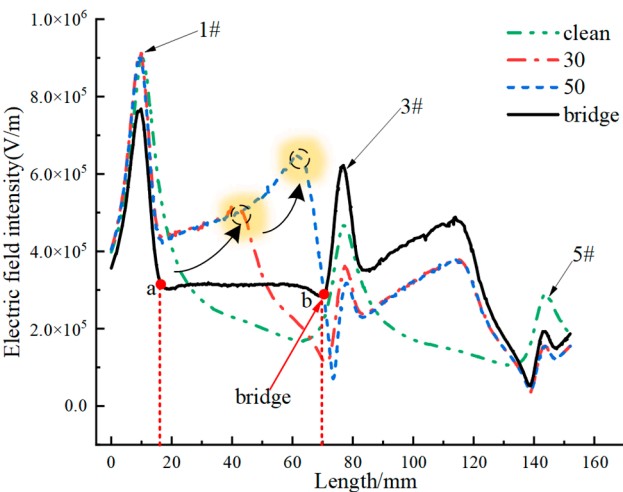

**Figure 14.** Trend of electric field strength before and after the ice ridge bridge connecting sheath.

Figure 14 shows that when the ice ridge is not bridged the sheaths, with the increase of the length of the ice ridge, the distortion of the space electric field is gradually strengthened, and the distortion reaches the maximum at the tip of the ice edge. It can be observed that an obvious peak exists in Figure 14. When the ice ridges bridge the sheaths, the bridged sheaths form an equipotential with the root of the ice edge. As shown in the section AB in Figure 14, the overall high potential of the insulator moves to the lower part, shortening the creepage distance of the insulator and increasing the probability of flashover to a great extent.

### 4.4. Influence of Ice Ridge Position on Electric Field Distribution

Maintaining the length of the ice ridges, change the positions where the ice ridges are attached, and enable them to be located at the edge of the sheath 1#, 3#, and 5#, respectively. Study the influence imposed by the positions of ice ridges on the distribution of electric field nearby. Take a three-dimensional line segment through the edges of sheath 1#, 3#, 5#

and 7#, with the starting point at the location 10 mm above the edge of sheath 1#, and the end point at the location 10 mm below the edge of sheath 7#, as shown in Figure 15. The line segment has a total length of 216 mm and the schematic diagram of its space is shown in Figure 15. Figure 16 is the cloud diagram about the distribution of electric field on the ice ridges with the same length on different sheaths. The distribution of electric field on the three-dimensional cutting line is shown in Figure 17.

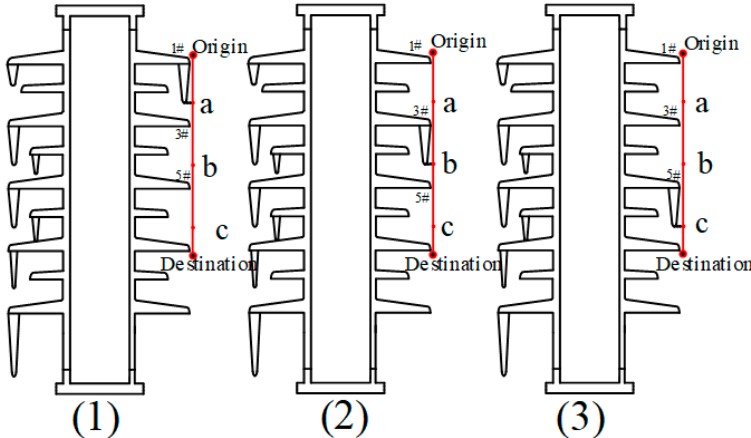

**Figure 15.** Ice ridge location and 3D section diagram: (**1**) The ice ridge is located on the 1# sheath; (**2**) The ice ridge is located on the 3# sheath; (**3**) The ice ridge is located on the 5# sheath.

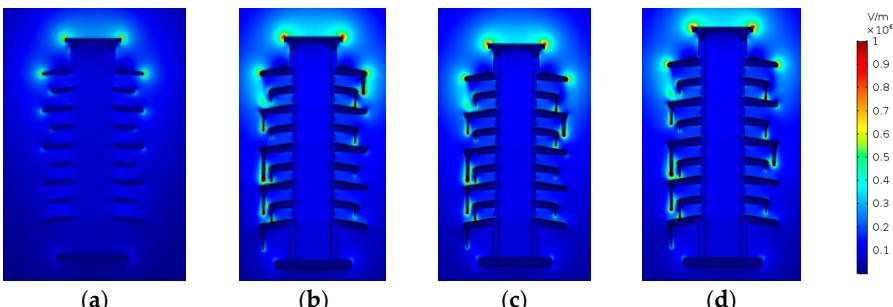

**Figure 16.** Cloud diagram of electric field intensity distribution at different positions of ice ridges. (**a**) Clean insulators; (**b**) Ice ridge at 1# sheath; (**c**) Ice ridge at 3# sheath; (**d**) Ice ridge at 5# sheath.

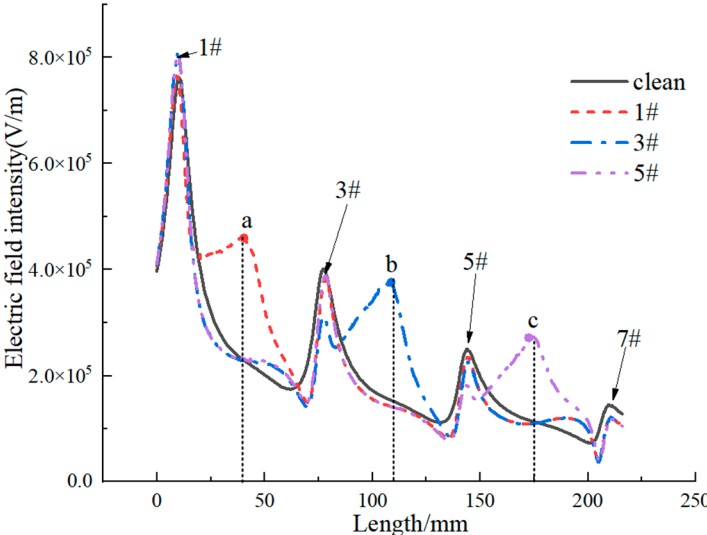

**Figure 17.** The variation trend of 3D cross-sectional electric field intensity.

It can be seen from Figures 16 and 17 that the distortion of the space electric field around the sheath of the insulator mostly occurs at the edges of the sheath under the condition that the insulator is not covered with ice. When the insulator is covered with ice, the distortion of the electric field is the most serious at the end of the ice ridges. Moreover, under the condition that the ice ridges have the same length, the closer it is to the end of high voltage, the more obvious the distortion of the electric field. As indicated in Figure 17, the relationship between points A, B and C in terms of electric field intensity is $E_a > E_b > E_c$.

### 4.5. Influence of Hanging Water Droplets at the End of Ice Ridge on Electric Field Distribution

During the process when the test of ice-covered flashover was conducted in Section 3.2, the ice layer on the surface of the sheath gradually melted, and the water of melted ice flowed down the ice ridges to form hanging water droplets on the tip of the ice ridges. With the aim of studying the influence imposed by the droplets of hanging water on the electric field in the surrounding space, a model of the hanging water droplets was established at the tip of the ice ridge, and the calculation of the electric field simulation was carried out. The cloud diagram of the electric field is shown in Figure 18.

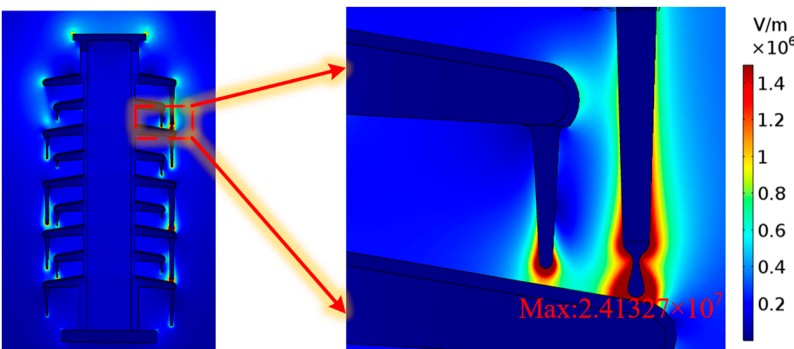

**Figure 18.** Cloud diagram of electric field intensity distribution of hanging water droplet.

It can be seen from Figure 18 that when the water droplets are hanging on the tip of the ice ridges, severe distortion is available in the space electric field around the water droplets, and the maximum value exists at the end of the water droplet, with the maximum value as high as $2.41 \times 10^7$ V/m. In addition, the conductivity shown by the water of melting ice is much greater than that of the dry ice ridge, and thus it provides the leakage of current with a path. When the suspended water droplets develop into a continuous ice melt flow, the arc of melting ice water bridges the sheaths, leading to the result of flashover easily.

### 5. Conclusions

- After the roof insulator is iced, the flashover path is the combination of "ice ridge–air gap" and sheath. The longer the ice ridge, the smaller the air gap. In the case of heavy icing, the ice ridge will bridge the sheath, which will reduce the creepage distance of the insulator, and the air gap between the tip of the ice ridge and the edge of the sheath will be easily broken down. In the structural design of the anti-icing insulator for the EMU, the spacing between adjacent sheaths and the extension of the sheaths should be considered to reduce the probability of ice ridges bridging the sheath.
- Ice layer and ice ridge can distort the electric field on the sheath surface and nearby. When the length of the ice ridge is 50 mm, the distortion rate of electric field intensity can reach as high as 5595% in the air gap seven millimeters below it, and under the same ice ridge length, the closer to the high-voltage end, the more obvious the distortion of the electric field. In a low temperature and high humidity environment, when the static ice ridge length of the roof insulator exceeds half the distance between adjacent sheaths, it is recommended to remove the ice in time to reduce the possibility of ice flashover failure.

- During the melting of the ice, there are hanging water droplets present at the ends of the ice ridge, and the electric field intensity at the ends of the water droplets is as high as $2.41 \times 10^7$ V/m., the water droplets hanging on the tip of an ice ridge are elongated by the joint action of electric field and gravity. The electric field distorted by the droplet and the continuous flow of ice melt water will provide the continued development of the arc with a shorter flashover path, quickening the pace for the formation of the final flashover channel.

**Author Contributions:** Conceptualization, J.M. and Z.Q.; methodology, J.M. and R.Z.; software, X.W. and L.M.; formal analysis, L.M. and Z.Q.; investigation, Z.Z. and Y.W.; writing—original draft preparation, Z.Q.; writing—review and editing, Z.Q.; funding acquisition, J.M. All authors have read and agreed to the published version of the manuscript.

**Funding:** This research was funded by National Nature Science Foundation of China (52167019); Gansu Natural Science Foundation (20JR5RA414) and Tianyou Innovation Team Science Foundation of Lanzhou Jiaotong University (TY202010).

**Institutional Review Board Statement:** Not applicable.

**Informed Consent Statement:** Not applicable.

**Data Availability Statement:** Not applicable.

**Conflicts of Interest:** The authors declare no conflict of interest.

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
