# Peer review of "Effects of Static Icing on Flashover Characteristics of High-Speed Train Roof Insulators"

_coatings, doi:10.3390/coatings12070950_

Round 1

Reviewer 1 Report

The authors analyze the electric field distribution for the insulator used in high speed railways.
The conducted analyzes do not bring any significant novelty in this respect.
The methods used are generally known and the results obtained are not sufficiently interpreted.
The authors do not draw conclusions as to the potential application of their results in practice.
In order for the work to be considered again, it is necessary to supplement it with further analyzes (studies).
Due to the title of the journal ("Coatings"), it is natural to carry out, for example, tests with the use of coatings that change the influence of icing on the field distribution.

Reviewer 2 Report

This paper described the flashover characteristics of icing insulator under the various icing structure. This paper includes useful information which seems to be interested by the many readers. The paper is well-organized and is based on original work. However, some descriptions in the text are not enough for clear understanding by the almost readers. Followings are comments for improvement of paper quality.

  1. Page 3, paragraph 1 from the bottom (lines 113-118): What is the frequency of the power supply used in the experiment? Please add the information.
  2. Page 5, lines 153-159: I guess that the surface of the ice during the experiment included a thin layer of water appeared by the melting of the ice. Could you add information about the resistance of the ice ridge during the experiment?
  3. Page 5, Figure 5: The size of photos is too small to confirm the location of discharges. Please modify the figure to confirm the phenomena explained in the text.
  4. Page 6, Figure 6(b): The photo in the area enclosed by the dotted circle has disappeared and the arc discharge phenomenon cannot be confirmed. Please modify the background of the dotted circle area change the color from white to transparent to confirm the discharge phenomenon.
  5. Pages 6-7, in 3.2 Ice Flashover test (lines 180-215): From Figs. 5-7 and its explanation, the overview of the discharge phenomenon for each phase can be confirmed. However, it is difficult to understand the details of each phenomenon (i.e., whether the discharge current was limited by the water resistance) owing to the lack of information about discharge current, discharge voltage and those waveforms (time-dependency) at each stage. Please add information on the voltage and current at each stage of the phenomenon.
  6. Pages 9-10, lines 289-307: Figure 14 and its explanation in the text can be understood easily and contain the useful information for the industry field. However, it seems to be no evidence based on the experiment. Can you provide experimental data to support this analysis? Experimental data on flashover or discharge ignition voltages for each structure would be useful to support the analysis.
  7. Pages 10-11, lines 309-332: Figure 17 and its explanation in the text contain the useful information. However, it seems to be no evidence based on the experiment. Can you also provide experimental data to support this analysis? The experimental data on voltages of flashover or discharge ignition is effective to support the analysis.

End.

Reviewer 3 Report

report is attached

Round 2

Reviewer 1 Report

Dear authors!

Paper is significantly improved.

But I have one more remark.

There must be information of potential practical implementation of results in conclusions.
